# Targeting brain lesions of non-small cell lung cancer by enhancing CCL2-mediated CAR-T cell migration

Hongxia Li[1,2], Emily B. Harrison [1,3], Huizhong Li[1,4], Koichi Hirabayashi[1], Jing Chen[5], Qi-Xiang Li [5], Jared Gunn[5], Jared Weiss[1,6,7], Barbara Savoldo[1,8], Joel S. Parker[1,9], Chad V. Pecot [1,6,7], Gianpietro Dotti [1,10✉] & Hongwei Du [1,4✉]

Metastatic non-small cell lung cancer (NSCLC) remains largely incurable and the prognosis is extremely poor once it spreads to the brain. In particular, in patients with brain metastases, the blood brain barrier (BBB) remains a significant obstacle for the biodistribution of anti-tumor drugs and immune cells. Here we report that chimeric antigen receptor (CAR) T cells targeting B7-H3 (B7-H3.CAR) exhibit antitumor activity in vitro against tumor cell lines and lung cancer organoids, and in vivo in xenotransplant models of orthotopic and metastatic NSCLC. The co-expression of the CCL2 receptor CCR2b in B7-H3.CAR-T cells, significantly improves their capability of passing the BBB, providing enhanced antitumor activity against brain tumor lesions. These findings indicate that leveraging T-cell chemotaxis through CCR2b co-expression represents a strategy to improve the efficacy of adoptive T-cell therapies in patients with solid tumors presenting with brain metastases.

[1] Lineberger Comprehensive Cancer Center, University of North Carolina at Chapel Hill, Chapel Hill, NC, USA. [2] Department of Medical Oncology, Beijing Chest Hospital, Capital Medical University, Beijing Tuberculosis and Thoracic Tumor Research Institute, Beijing, China. [3] Center for Nanotechnology in Drug Delivery, Eshelman School of Pharmacy, University of North Carolina at Chapel Hill, Chapel Hill, NC, USA. [4] Cancer Immunotherapy Center, Cancer Research Institute, Xuzhou Medical University, Xuzhou, Jiangsu Province, China. [5] Crown Bioscience Inc, San Diego, CA, USA. [6] Division of Hematology/Oncology, University of North Carolina at Chapel Hill, Chapel Hill, NC, USA. [7] Department of Medicine, University of North Carolina at Chapel Hill, Chapel Hill, NC, USA. [8] Department of Pediatrics, University of North Carolina at Chapel Hill, Chapel Hill, NC, USA. [9] Department of Genetics, University of North Carolina at Chapel Hill, Chapel Hill, NC, USA. [10] Department of Microbiology and Immunology, University of North Carolina at Chapel Hill, Chapel Hill, NC, USA. ✉email: gdotti@med.unc.edu; hwdu@xzhmu.edu.cn

Non-small cell lung cancer (NSCLC) accounts for nearly 85% of all cases of lung cancer, and remains a fatal disease for the great majority of patients[1]. Conventional radiotherapy, chemotherapy, and targeted therapies including immune checkpoint inhibitors have improved the survival of NSCLC patients in the past decades, but the 5-year survival rate of patients with metastatic disease remains poor[2,3]. Furthermore, approximately 10% of newly diagnosed patients present with brain metastasis[4], and 25–40% of patients develop brain metastases after the initial diagnosis[5,6]. Although some newer therapeutics cross the BBB, and are clinically active against NSCLC brain metastases[7,8], the BBB remains a significant physical obstacle to drug biodistribution[9]. Thus, new therapeutic strategies are urgently needed for NSCLC patients, especially for those with brain metastases.

Immunotherapy with chimeric antigen receptor (CAR) engineered T cells has achieved remarkable success in hematologic malignancies[10–13]. In contrast, CAR-T cells have only shown modest clinical activity in solid tumors thus far. Limited availability of targetable antigens allowing selective elimination of the tumor cells, heterogeneity in antigen expression in tumor cells, and suboptimal trafficking of CAR-T cells outside the lymphoid organs account at least in part for the limited activity of CAR-T cells in solid tumors[14]. Furthermore, as is true for chemotherapies, small molecule inhibitors, and monoclonal antibodies (mAbs), brain metastases represent the most difficult location to be reached by adoptively transferred CAR-T cells[15–17].

Here, we propose a combination of optimized tumor recognition and T-cell trafficking that controls multiple locations of NSCLC, including brain metastases. Specifically, we find significant antitumor activity against orthotopic and metastatic NCSLC, including brain localizations, when combining B7-H3 targeting via CAR-T cells with the engineering of the CCL2/CCR2 chemokine/chemokine receptor axis.

## Results

**B7-H3 is expressed in NSCLC and is targeted by B7-H3.CAR-T cells in vitro and in organoid models.** Frozen human NSCLC tissue microarrays (TMA) were stained with the B7-H3 mAb 376.96. All NSCLC specimens, which included both adenocarcinoma and squamous cell carcinomas, stained positive for B7-H3 expression (Fig. 1a, b). We have generated a CAR targeting B7-H3 (B7-H3.CAR) using the single-chain variable fragment (scFv) derived from the mAb 376.96 that includes either CD28 (B7-H3.28) or 4-1BB (B7-H3.BB) endodomains for costimulation (Supplementary Fig. 1a–c)[18]. To evaluate the antitumor activity of B7-H3.CAR-T cells against human NSCLC, five B7-H3+ human NSCLC cell lines (A549, H1299, H460, H520, and SK-MES-1) were cocultured with control CD19.28, B7-H3.28 or B7-H3.BB CAR-T cells at T cell to tumor cell ratio of 1:5 (Fig. 1c). We found that B7-H3.28 and B7-H3.BB CAR-T cells effectively controlled NSCLC cell growth (Fig. 1d, e), and their cytolytic activity was corroborated by IFNγ and IL2 release in the culture supernatant (Fig. 1f, g), and by T cell proliferation in response to B7-H3+ target cells, as assessed by CFSE dilution assay (Supplementary Fig. 1d). To further analyze the antitumor effects of B7-H3-specific CAR-T cells in a model system using primary tumor cells, we implemented coculture experiments using a lung cancer organoid (Fig. 1h) [https://organoid.crownbio.com]. When the B7-H3+ organoid LU6438B was cocultured with CD19.28 or B7-H3.28 CAR-T cells at T cell to organoid cell ratios of 1:2 and 1:5, we found that B7-H3.28 CAR-T cells effectively eliminated B7-H3+ cells accompanied with IFNγ and IL2 secretion (Fig. 1i–l and Supplementary Fig. 1e). Taken together, these data indicate that B7-H3 is a suitable target for CAR-T cells in NSCLC.

**B7-H3.CAR-T cells show antitumor activity in metastatic and orthotopic NSCLC xenograft models.** To investigate the antitumor effects of B7-H3.CAR-T cells in vivo, we implemented a metastatic NSCLC model by infusing FFluc-expressing A549 tumor cells via tail vein injection (i.v.) in NSG mice, and treated them with CAR-T cells (Fig. 2a). Both B7-H3.28 and B7-H3.BB CAR-T cells effectively controlled A549 tumor growth, and the mice remained tumor free up to 98 days post treatment (Fig. 2b–d). Similar results were obtained in mice engrafted with the FFluc-expressing H520 tumor cells (Supplementary Fig. 2a–d). We also established an orthotopic NSCLC model by implanting FFluc-A549 tumor cells into the left lung of NSG mice. Seven days later, mice were infused i.v. with CAR-T cells (Fig. 2e). B7-H3.28 and B7-H3.BB CAR-T cells effectively controlled tumor growth as compared to control CD19.28 CAR-T cells (Fig. 2f–h). Overall, B7-H3.28 and B7-H3.BB CAR-T cells exhibited equal antitumor effects in NSCLC metastatic and orthotopic models in vivo. However, when tumor cells were engrafted intracranially to mimic the scenario of brain tumor localization in NSCLC, B7-H3.CAR-T cells infused i.v. did not effectively control the tumor growth, even when high-dose CAR-T cells were used (Supplementary Fig. 3). These data indicate that although B7-H3.CAR-T cells effectively controlled NSCLC tumor growth in orthotopic and metastatic models, they cannot eradicate tumor cells located in the brain, which may be due to the BBB hampering CAR-T cell trafficking.

**CCL2 is highly expressed in NSCLC in primary tumors and brain lesions and attracts CAR-T cells expressing CCR2b.** Chemokine gradients established by tumor cells can attract immune cells within the tumor microenvironment, and CAR-T cells can be engineered to enhance their migration toward tumors by exploiting specific chemokine gradients[19]. We further hypothesized that chemokine gradients could also be exploited to overcome the physical obstacle of the BBB. Therefore, we investigated the chemokine gene expression profile of NSCLC from the TCGA data set. From the analysis of 41 CCL and CXCL genes, we found that CCL2 was one of the most highly expressed genes among all chemokine family members. Expression of CCL2 was >88% and >89% of all other chemokine family members in the TCGA lung cancer adenocarcinoma (LUAD) and lung cancer squamous cell carcinoma (LUSC), respectively (Fig. 3a and Supplementary Table 1). Sequencing data were corroborated by the detection of significant amounts of CCL2 in the supernatant from several cultured NSCLC cell lines (Fig. 3b). In addition, analysis of the chemokine expression profiles of NSCLC brain metastases in the data set generated by Kudo et al.[20] indicated that in 37 subjects with primary NSCLC and corresponding brain metastasis, CCL2 was more expressed in the brain metastases compared with the primary lung lesions. Furthermore, CCL2 was amongst the top 10% across 39 chemokine family members in the paired brain metastasis samples (Fig. 3c and Supplementary Table 1). Based on this evidence, we hypothesized that leveraging the CCL2/CCR2 chemokine/chemokine receptor axis may promote chemo-attraction of CAR-T cells to brain metastatic lesions. Because CCR2 is only marginally expressed on T cells, we engineered T cells to overexpress either CCR2a or CCR2b, which are the two isoforms of the human CCL2 receptor, and then tested their migration properties toward CCL2. We found that CCR2b expressed at higher and more stable levels and caused superior migration toward the CCL2 gradient than CCR2a. Thus, CCR2b was selected for further investigation (Supplementary Fig. 4a–c). We constructed bicistronic vectors encoding the B7-H3.CAR and CCR2b (Fig. 3d). CCR2 expression was significantly increased in CCR2b.B7-H3.28 and CCR2b.B7-H3.BB CAR-T cells compared

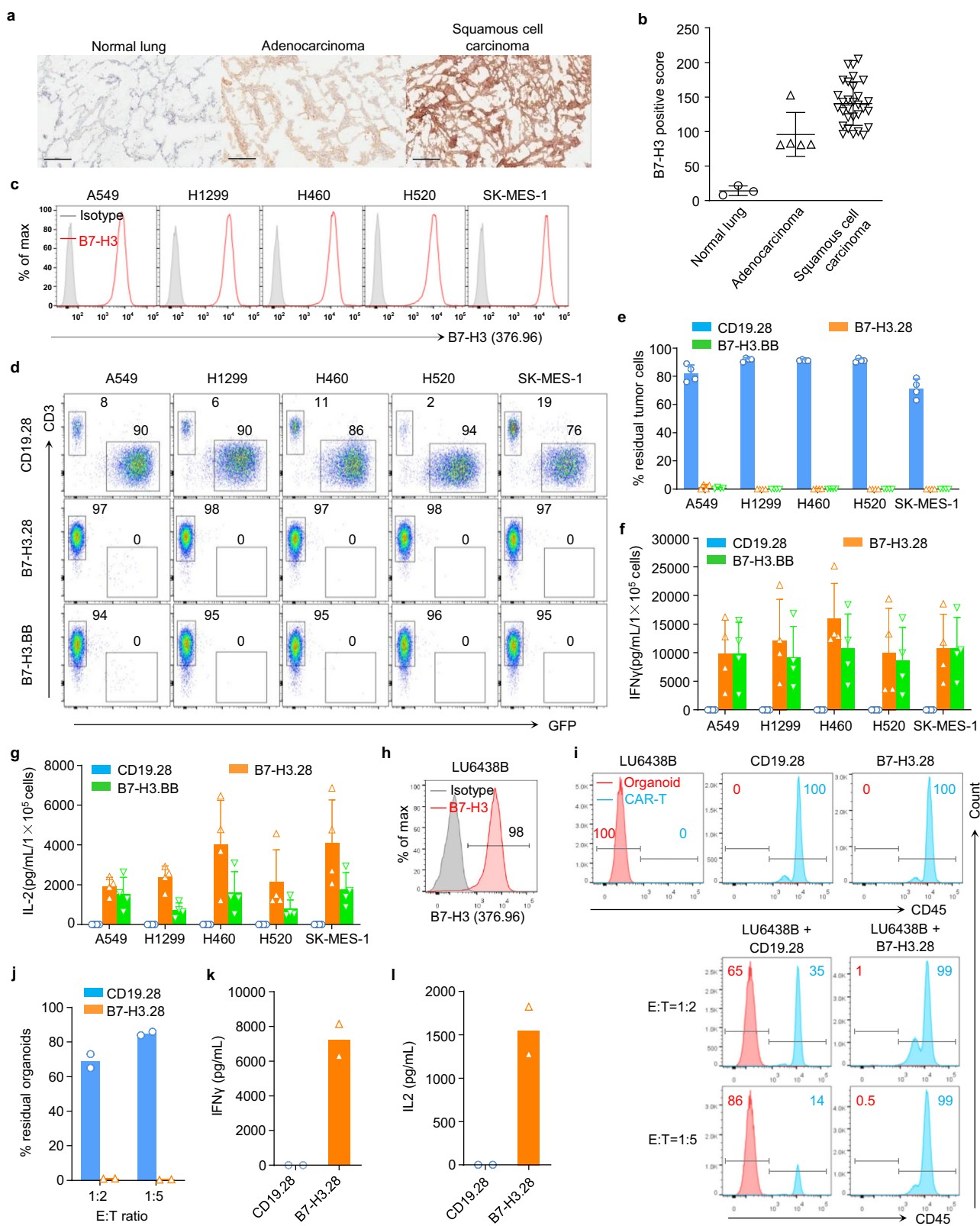

to the control non-transduced T cells (NT) and B7-H3.28 or B7-H3.BB CAR-T cells (Fig. 3e, f), while these cells retained similar expansion rate and phenotypic composition of central-memory, effector-memory, and T stem cell memory (Supplementary Fig. 5a–d). CCR2b overexpression conferred increased migration of CCR2b.B7-H3.28 and CCR2b.B7-H3.BB CAR-T cells toward

recombinant CCL2 and supernatant collected from NSCLC cell lines (Fig. 3g). Overall, these data underline the clinical relevance of the CCL2 gradient in primary lesions and brain metastases of NSCLC and that the CCL2/CCR2 axis can be genetically manipulated to promote CAR-T cell migration toward the CCL2 gradient.

**Fig. 1 B7-H3.CAR-T cells target B7-H3$^+$ NSCLC cell lines and organoid in vitro. a** Representative micrographs showing B7-H3 expression in cryosections of normal lung and NSCLC assessed by staining with the 376.96 mAb at the final concentration of 1 μg/mL. Scale bars, 200 μm. **b** B7-H3 expression score summary of the immunochemistry results in normal lung and NSCLC showing in (**a**), $n = 3$ for normal lung, $n = 5$ for adenocarcinoma, $n = 30$ for squamous cell carcinoma. Data are presented as mean values ± SD. **c** Representative flow cytometry plots showing B7-H3 expression in NSCLC cell lines, stained with the 376.96 mAb antibody. **d**, **e** NSCLC cell lines labeled with GFP were cocultured with CD19.28, B7-H3.28, or B7-H3.BB CAR-T cells at the T cell to tumor cell ratio of 1:5. On day 5, NSCLC cells (GFP$^+$) and CAR-T cells (CD3$^+$) were enumerated by flow cytometry. Representative flow-cytometry plots (**d**) and quantification of residual tumor cells (**e**) are illustrated. Data are presented as mean values + SD. **f**, **g** Summary of IFNγ (**f**) and IL2 (**g**) released by CAR-T cells in the culture supernatant after 24 h of coculture with the indicated cell lines as measured by ELISA, $n = 4$ independent experiments using CAR-T cells generated from four different donors in (**d–g**). Data are presented as mean values + SD. **h** Representative flow plots showing B7-H3 expression in lung cancer organoid LU6438B, which was stained with the 376.96 mAb antibody. **i**, **j** Organoid LU6438B was cocultured with CD19.28 or B7-H3.28 CAR-T cells at the T cell to organoid cell ratio of 1:2 and 1:5. On day 4, organoid cells (CD45$^-$) and CAR-T cells (CD45$^+$) were identified by flow cytometry. Representative flow-cytometry plots (**i**) and quantification of residual organoid cells (**j**) are illustrated. Data are presented as mean values. **k**, **l** Summary of IFNγ (**k**) and IL2 (**l**) released by CAR-T cells in the supernatant after 24 h of coculture with organoids as measured by ELISA. Data are presented as mean values. $N = 2$ independent experiments using CAR-T cells generated from two different donors. Source data for (**b**, **e**, **f**, **g**, **j**, **k**, **l**) are provided as a Source Data file.

**CCR2b co-expressing B7-H3.CAR-T cells have superior anti-tumor activity against NSCLC brain lesions.** CCR2b coexpression with the CAR slightly decreased the MFI of CAR expression without affecting the overall transduction efficiency of the CAR-T cells (Fig. 4a–c). To evaluate the antitumor activity of CCR2b engineered B7-H3.CAR-T cells, we cocultured them with tumor cells and found that CCR2b coexpression did not hamper their antitumor activity, cytokine release, and proliferative capacity in vitro (Fig. 4d–g). We then tested the effects of CCR2b over-expression in B7-H3.CAR-T cells in vivo in both metastatic models and brain tumor models of FFluc-A549 and FFluc-H520 cells. For these experiments, we selected the B7-H3.28 CAR as proof of concept. In the metastatic model of NSCLC, CCR2b.B7-H3.28 CAR-T cells showed slightly enhanced antitumor activity compared to B7-H3.28 CAR-T cells (Supplementary Fig. 6a–f). In sharp contrast, in the NSCLC brain tumor model, CCR2b.B7-H3.28 CAR-T cells administered i.v. were significantly more effective than B7-H3.28 CAR-T cells. Seven out of nine mice (78%) treated with CCR2b.B7-H3.28 CAR-T cells remained tumor free up to 98 days post treatment (Fig. 5a–d). In separate experiments, in which mice were sacrificed at day 8 post CAR-T cell infusion for enumerating CAR-T cells in blood, spleen and tumor, we found that T-cell numbers in blood and spleen were similar in mice treated with B7-H3.28 or CCR2b.B7-H3.28 CAR-T cells, while we observed a significant increase of CAR-T cells in the tumor in mice receiving CCR2b.B7-H3.28 CAR-T cells (Fig. 5e). To further investigate if the increased infiltration of CCR2b.B7-H3.CAR-T is tumor specific, we conducted another experiment, in which the mice were also sacrificed on day 8 post CAR-T cell infusion to further enumerate CAR-T cells in the left brain hemisphere (without tumor) and right brain hemisphere (with tumor). We found that T-cell numbers in the left brain hemisphere were similar in mice treated with B7-H3.28 or CCR2b.B7-H3.28 CAR-T cells, while we observed a significant increase of CAR-T cells in the right brain hemispheres of mice treated with CCR2b.B7-H3.28 CAR-T cells compared to the mice treated with B7-H3.28 CAR-T cells (Fig. 5f). These data indicate that the enhanced infiltration of CCR2b.B7-H3.28 CAR-T cells is tumor specific. In addition, we also found that CCR2b.B7-H3.28 CAR-T cells exhibited superior antitumor activity in a second NSCLC brain tumor model, in which the mice were engrafted i.c. with FFluc-H520 tumor cell. Consistently, we observed that the CCR2b.B7-H3.28 CAR-T cells were more effective than B7-H3.28 CAR-T cells in controlling H520 tumor growth (Supplementary Fig. 7a–d). In this model, we also observed significantly increased numbers of CAR-T cells in the brain hemisphere with tumor in mice treated with CCR2b.B7-H3.28 compared to B7-H3.28 CAR-T cells (Supplementary Fig. 7e).

To explore the effect of CCR2b expression on CAR-T cells when tumor lesions are present simultaneously in the lung and in the brain, we implemented a model in which NSG mice are engrafted with tumor cells in both lung and brain using two sequential surgery procedures. When these mice were treated with CD19.28, B7-H3.28, or CCR2b.B7-H3.28 CAR-T cells, we found that CCR2b.B7-H3.28 CAR-T cells continued showing superior antitumor activity against the tumor lesions in the brain (Fig. 6a–d). Overall, these data indicate that CAR-T cells expressing CCR2b provide superior control of the tumor localized within the brain compared to T cells expressing the CAR alone.

## Discussion

In this study, we found that B7-H3.CAR-T cells effectively era-dicated NSCLC tumor cell lines and organoids in vitro and controlled tumor growth in xenotransplant models. More importantly, we found that the CCL2 gradient detected in primary NSCLC as well as in NSCLC brain metastases can be exploited to promote efficient migration of B7-H3.CAR-T cells through the BBB and to enhance antitumor effects when these cells overexpress CCR2b.

Mesothelin, human epidermal growth factor receptor 2, epi-dermal growth factor receptor, and more recently the GD2 ganglioside have been proposed as candidates for CAR-T cell targeting in NSCLC[21–24]. Here we found that B7-H3 is another suitable candidate to target NSCLC via CAR-T cells. Antitumor activity of B7-H3.CAR-T cells was found not only by using NSCLC tumor cell lines, but also by using a patient-derived xenograft NSCLC organoid model that represents a more clini-cally relevant surrogate in vitro of primary NSCLC lesions[25–27]. Antitumor activity was also observed in both metastatic and orthotopic xenotransplant models without any significant differ-ences between CD28 and 4-1BB costimulation supporting the translational potential of B7-H3.CAR-T cells in NSCLC. The clinical translation of B7-H3.CAR-T cells in NSCLC is further encouraged by reported data indicating that B7-H3 is highly expressed in multiple tumors, while the expression in normal tissues is low or absent[18,28–31]. Furthermore, a recent case report in which B7-H3.CAR-T cells were infused in a patient with meningioma did not report detectable toxicity[32].

Despite controlling both orthotopic and metastatic tumors, B7-H3.CAR-T cells inoculated intravenously were inefficient in controlling NSCLC engrafted within the brain. Intratumor or intraventricular delivery of CAR-T cells is actively investigated in patients with primary brain tumors due to their exclusive loca-lization within the brain[15,33]. In contrast, brain metastases of solid tumors such as NSCLC and other solid tumors may benefit

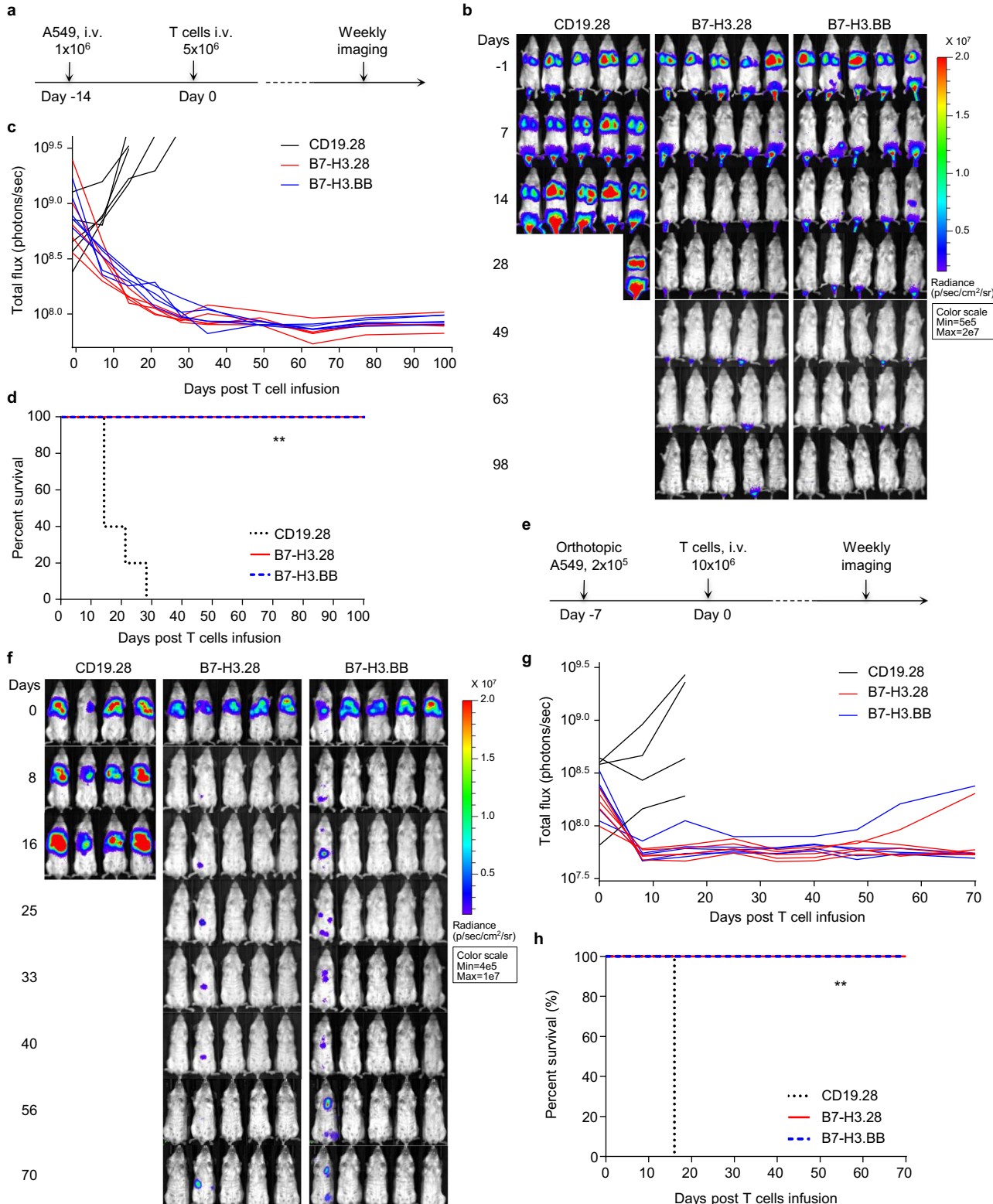

**Fig. 2 B7-H3.CAR-T cells eradicate B7-H3+ NSCLC in metastatic *and* orthotopic models. a** Schematic representation of a metastatic NSCLC model in NSG mice using the FFluc-A549 cell line. Representative images of tumor bioluminescence (BLI) (**b**) and kinetics (**c**) of tumor growth ($n = 5$ mice/group). **d** Kaplan–Meier survival curve of mice in (**b**) ($n = 5$ mice/group), $^{**}p = 0.0015$ (B7-H3.28 vs. CD19.28 CAR-T cells), $^{**}p = 0.0015$ (B7-H3.BB vs. CD19.28 CAR-T cells) $\chi^2$ test. In this model for the survival curve, mice were censored when the luciferase signal reached $3.5 \times 10^9$ photons per second. **e** Schematic representation of an orthotopic NSCLC model in NSG mice using the FFluc-A549 cell line. Representative images of tumor BLI (**f**) and kinetics (**g**) of tumor growth ($n = 5$ mice/group). **h** Kaplan–Meier survival curve of mice in (**e**) ($n = 5$ mice/group), $^{**}p = 0.0047$ (B7-H3.28 vs. CD19.28 CAR-T cells), $^{**}p = 0.0047$ (B7-H3.BB vs. CD19.28 CAR-T cells) $\chi^2$ test. In this model for the survival curve, mice were censored when the luciferase signal reached $3.5 \times 10^9$ photons per second. Days indicated in (**b** and **f**) are days post T cell infusion. Source data for (**c, d, g, h**) are provided as a Source Data file.

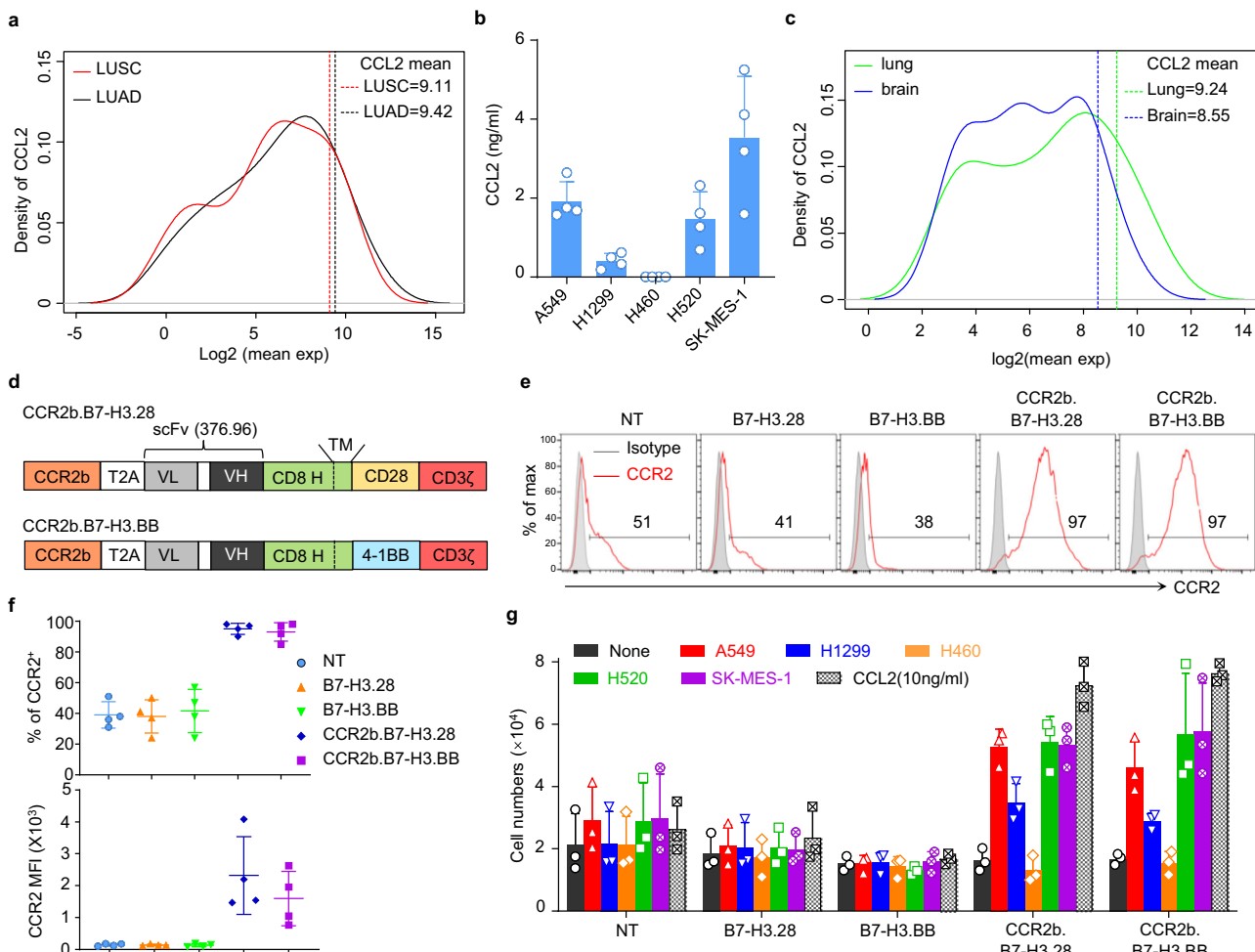

**Fig. 3 Coexpression of CCR2b in B7-H3.CAR-T cells improves the migration toward CCL2. a** Mean gene expression distributions of CCL and CXCL family members (41 genes) in lung cancer adenocarcinoma (LUAD) ($n = 517$) and lung cancer squamous cell carcinoma (LUSC) ($n = 501$) cohorts from TCGA data set. The solid curve lines indicate the distribution of the quantiles of CCL2 in all 41 chemokines genes at each expression level, and the dotted line indicates the mean expression of CCL2 in LUAD and LUSC. **b** CCL2 measured in the culture supernatant of NSCLC cell lines ($1 \times 10^5$/mL cells cultured in 24-well plates for 24 h) ($n = 4$ independent samples). Data are presented as mean values + SD. **c** Mean gene expression distributions of CCL and CXCL family members (39 genes) from 37 NSCLC paired with brain metastasis. The solid curve lines indicate the distribution of the quantiles of CCL2 in all the 39 genes at each expression level and the dotted line indicates the mean expression of CCL2 in lung cancer and brain metastasis. CCL2 expression is in the top 15 and 8% quantiles of the 39 CCL and CXCL genes expression in primary lung cancer tissue and brain metastasis, respectively. **d** Schematic of the vectors encoding B7-H3.CAR and CCR2b. **e** Representative flow plots showing CCR2 expression in control non-transduced T cells (NT), B7-H3.28, B7-H3.BB, CCR2b.B7-H3.28 and CCR2b.B7-H3.BB CAR-T cells. **f** Summary of CCR2 expression percentage (top) and MFI (bottom) in NT, B7-H3.28, B7-H3.BB, CCR2b.B7-H3.28 and CCR2b.B7-H3.BB CAR-T cells ($n = 4$ independent experiments using CAR-T cells generated from four different donors). Data are presented as mean values ± SD. **g** Cell numbers of NT, B7-H3.28, B7-H3.BB, CCR2b.B7-H3.28 and CCR2b.B7-H3.BB CAR-T cells migrated to the bottom of the transwell chambers with 5 μm pores that were induced by the culture supernatant of NSCLC cell lines and human CCL2 recombinant protein (10 ng/mL), ($n = 3$ independent experiments using CAR-T cells generated from three different donors). Data are presented as mean values + SD. Source data for (**b**, **f**, **g**) are provided as a Source Data file. Source data for (**a**, **c**) are provided as Supplementary Table 1.

from the intravenous administration of CAR-T cells because brain metastases are generally multifocal and develop concomitantly with the presence of other metastatic lesions in the body. Furthermore, CAR-T cells administered intravenously in patients with glioblastoma demonstrated a suboptimal accumulation within the tumor, which can be attributed at least in part to the presence of the BBB[15–17].

Models of experimental autoimmune encephalomyelitis (EAE) provided evidence that pro-inflammatory chemokines such as CCL2 produced mostly by astrocytes are involved in the pathogenesis of EAE and recruit T cells[34]. Here we provide evidence that CCL2 is detected in both primary tumors and brain metastases of human NSCLC cells, and that NSCLC cell lines secrete CCL2, which attracts B7-H3.CAR-T cells when these cells are

engineered to overexpress CCR2b. RNA sequence data showed that CXCL1 and CXCL2 are the highest chemokine genes expressed in brain metastatic lesions. However, CXCL1/CXCL2 corresponding receptor CXCR2 binds to multiple chemokines including CXCL3, CXCL5, CXCL6, CXCL7, CXCL8, acPGP, and MIF[35]. It has been shown that CXCR2 overexpression can enhance CAR-T cell's antitumor activity, but since CXCR2 expressing CAR-T cells may respond to multiple chemokines and acquire wide migration to multiple normal tissues, thus there is a concern that this approach may cause toxicity[36–38].

We have previously reported that manipulating the TARC/CCR4 chemokine/chemokine receptor pathway enhances CAR-T cell antitumor activity in Hodgkin's lymphoma models[39]. CAR-T cells overexpressing CCR2 has been found to enhance the

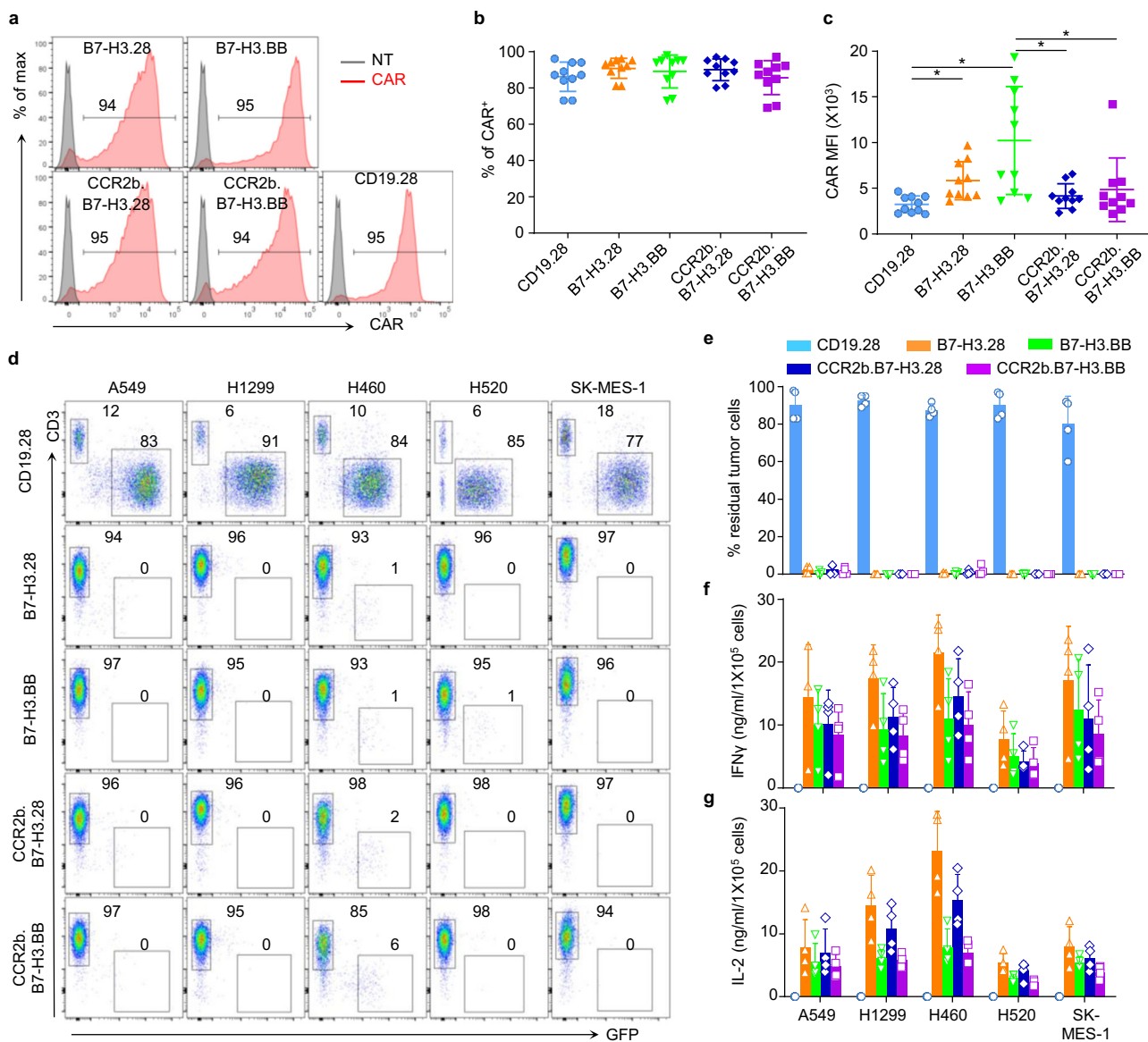

**Fig. 4 CCR2b co-expressing B7-H3.CAR-T cells have comparable antitumor activity when coculture with NSCLC cell lines in vitro. a** Representative flow plots of CARs expression in CD19.28, B7-H3.28, B7-H3.BB, CCR2b.B7-H3.28 and CCR2b.B7-H3.BB CAR-T cells as assessed by flow cytometry. **b, c** Summary of the percentage (**b**) and MFI (**c**) of CAR expression in B7-H3.28, B7-H3.BB, CCR2b.B7-H3.28 and CCR2b.B7-H3.BB CAR-T cells ($n = 10$ independent experiments using CAR-T cells generated from ten different donors, $*p = 0.0219$ for CD19.28 vs. B7-H3.28, $*p = 0.0240$ for CD19.28 vs. B7-H3.BB, $*p = 0.0295$ for B7-H3.BB vs. CCR2b.B7-H3.28, $*p = 0.0217$ for B7-H3.BB vs. CCR2b.B7-H3.BB, one-way ANOVA). Data are presented as mean values ± SD. **d, e** GFP-labeled NSCLC cell lines were cocultured with CD19.28, B7-H3.28, B7-H3.BB, CCR2b.B7-H3.28 or CCR2b.B7-H3.BB CAR-T cells at the CAR-T cell to tumor cell ratio of 1:5. On day 5, NSCLC cells (GFP+) and CAR-T cells (CD3+) were enumerated by flow cytometry. Representative flow-cytometry plots (**d**) and quantification of residual tumor cells (**e**) are illustrated. Data are presented as mean values + SD. **f, g** Summary of IFNγ (**f**) and IL2 (**g**) released by CAR-T cells in the culture supernatant after 24 h of coculture with the indicated cell lines as measured by ELISA, $n = 4$ independent experiments using CAR-T cells generated from four different donors in (**d**–**g**). Data are presented as mean values + SD. Source data for (**b**, **c**, **e**, **f**, **g**) are provided as a Source Data file.

migration and antitumor activity of CAR-T cells in xenograft neuroblastoma and mesothelioma models[40,41]. However, whether manipulating chemokine/chemokine receptor pathways can drive CAR-T cells through the BBB has not been investigated. Here we demonstrated that coexpression of CCR2b only modestly improved the antitumor efficacy of B7-H3.CAR-T cells in metastatic xenograft tumor models of NSCLC. In this NSCLC metastatic model, tumor cells engraft mostly in the liver and the chemokine gradients may play a limited role in governing the trafficking of CAR-T cells to the liver when the CAR-T cells are infused intravenously. In contrast, CCR2b coexpression in B7-

H3.CAR-T cells significantly enhanced the infiltration and accumulation of CAR-T cells in tumor models in which tumors are localized in the brain, and therefore causing sustained regression of these lesions. Of note, in the tumor model in which NSCLC tumor cells are engrafted both in the lung and in the brain, the CCR2b coexpression in B7-H3.CAR-T cells continued to show activity against the tumor in the brain suggesting that CCR2b.B7-H3.CAR-T cells retain enhanced trafficking through the BBB even if the primary tumor is still present in the lung. Finally, we observed that the CCL2/CCR2-mediated trafficking of CCR2b.B7-H3.CAR-T cells to the tumor within the brain is

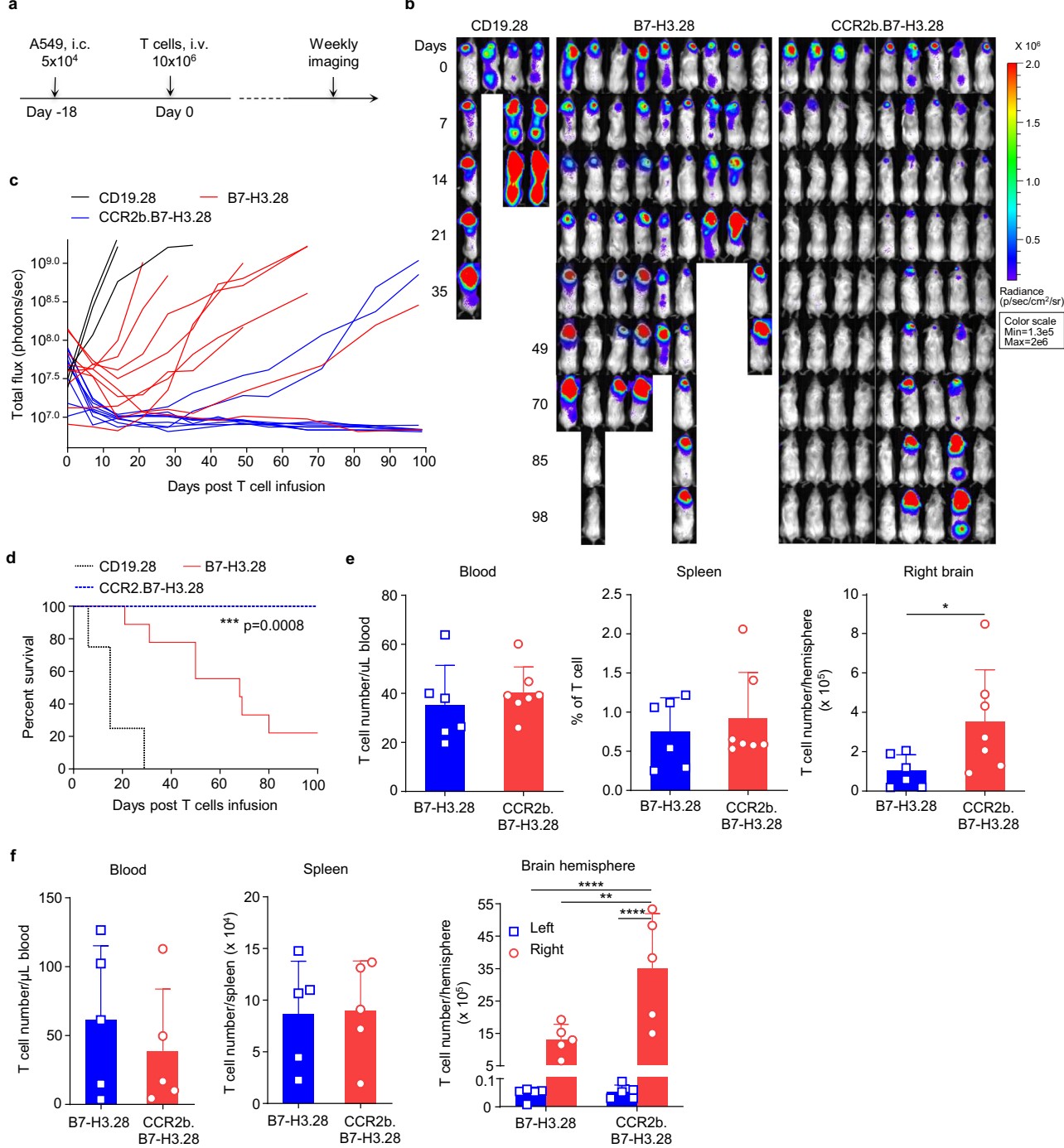

**Fig. 5 B7-H3.CAR-T cells co-expressing CCR2b have superior antitumor activity against the A549 tumor in the brain. a** Schematic of the A549 brain tumor model in which NSG mice were implanted intracranially (i.c.) with FFluc-A549 cells ($5 \times 10^4$ cells) into the right brain hemisphere, and then treated with CD19.28, B7-H3.28, or CCR2b.B7-H3.28 CAR-T cells ($10 \times 10^6$) inoculated i.v. 18 days later. **b, c** Representative tumor BLI images (**b**) and BLI kinetics (**c**) in the model shown in (**a**). Days indicated in (**b**) represent the days post T cell infusion. **d** Kaplan–Meier survival curves of mice in (**b**). ***$p = 0.0008$ (CCR2b.B7-H3.28 vs. B7-H3.28), $\chi^2$ test. $N = 4$ mice in CD19.28 group, and $n = 9$ mice in B7-H3.28 and CCR2b.B7-H3.28 in (**c, d**). **e** In the separate experiment, mice were euthanized 8 days after CAR-T cell infusion, and human CD45+CD3+ T cells were enumerated in blood, spleen, and the right brain hemisphere by flow cytometry. Bar graph summary are shown, data are presented as mean values + SD ($n = 6$ mice in B7-H3.28, $n = 7$ mice in CCR2b.B7-H3.28), *$p = 0.0464$, unpaired $t$ test with Welch's correction and two-tailed $p$ value calculation. **f** In another experiment, the same procedure as the experiment in (**e**), when the mice were euthanized 8 days after CAR-T cell infusion, human CD45+CD3+ T cells in both the left brain hemisphere (without tumor) and right brain hemisphere (with tumor) were enumerated by flow cytometry, besides blood and spleen. Bar graph summary are shown, data are presented as mean values + SD ($n = 5$ mice/group), **$p = 0.0062$, ****$p < 0.0001$, two-way ANOVA with Sidak's multiple comparisons test. Source data for (**c, d, e, f**) are provided as a Source Data file.

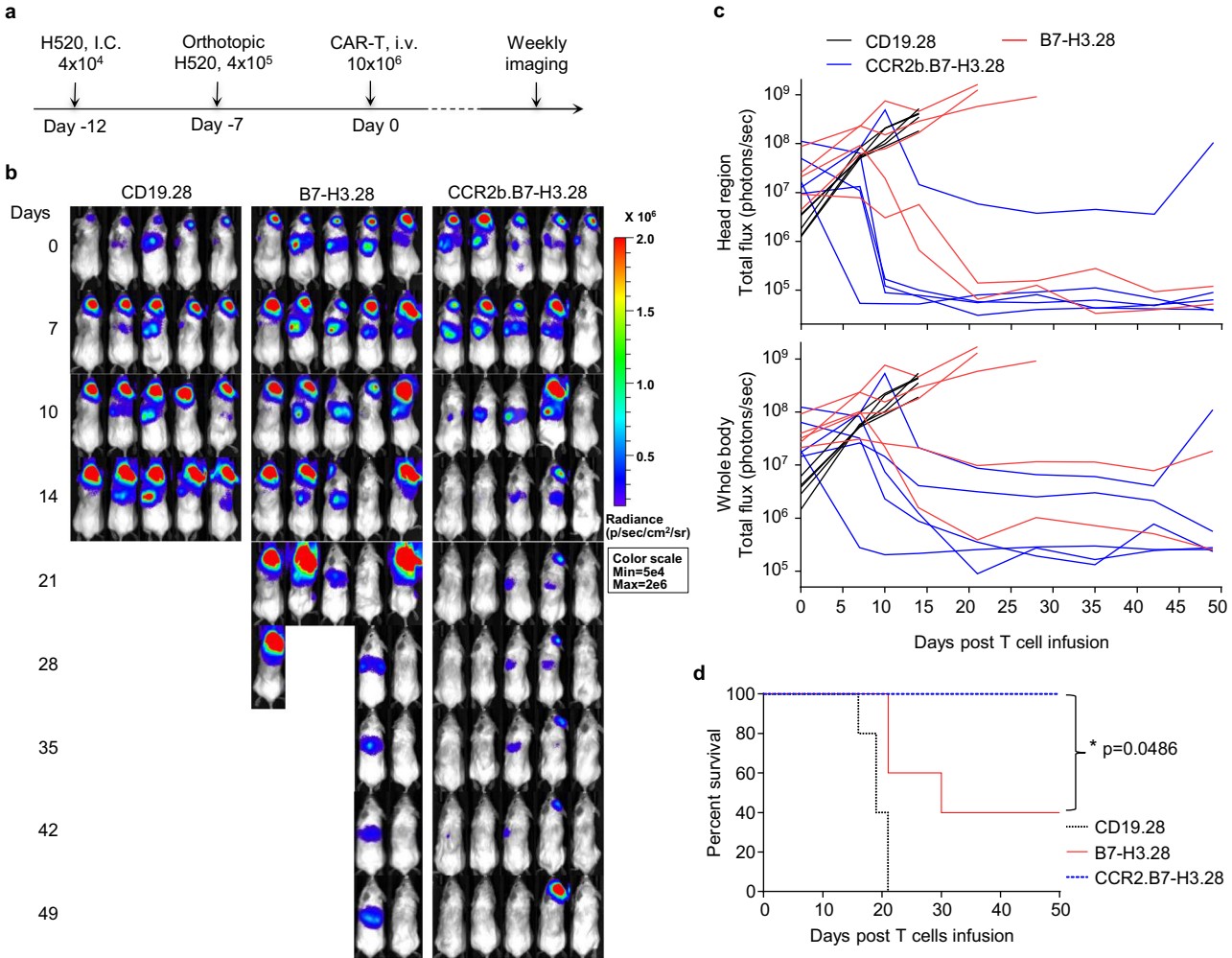

**Fig. 6 CCR2b co-expressing B7-H3.CAR-T cells show superior antitumor activity in the H520 lung orthotopic and brain co-xenograft tumor model.** **a** Schematic of the H520 lung orthotopic and brain co-xenograft tumor model in NSG mice. Briefly, the FFluc-H520 ($4 \times 10^4$) cells were implanted into the right hemisphere of the brain of NSG mice by i.c. injection. Five days later, $4 \times 10^5$ FFluc-H520 cells were implanted into the left lung by intrathoracic injection. Seven days later, mice were treated with CD19.28, B7-H3.28, or CCR2b.B7-H3.28 CAR-T cells ($10 \times 10^6$ cells/mouse) by i.v. injection. **b** Representative tumor BLI images. Days represent the days post T-cell infusion. **c** The BLI kinetics of the FFluc-H520 tumor growth in the model shown in (**b**), BLI signals from the brain (top) and whole body (bottom) are illustrated, respectively ($n = 5$ mice/group). **d** Kaplan–Meier survival curves of mice in (**b**), $n = 5$ mice/group, *$p = 0.0486$ for B7-H3.28 vs. CCR2b.B7-H3.28, *$p = 0.0181$ for CD19.28 vs. B7-H3.28, **$p = 0.0026$ for CD19.28 vs. CCR2b.B7-H3.28, $\chi^2$ test. Source data for (**c**, **d**) are provided as a Source Data file.

tumor specific since we did not observe accumulation of CCR2b.B7-H3.CAR-T cells in the opposite hemisphere of the brain.

Collectively, we demonstrated that CCR2b coexpression allows for significantly enhanced eradication of brain metastatic lesions of NSCLC. The concept of tailoring chemokine receptor to match chemokines expressed in brain metastases provides a unique approach to control these lethal tumor localizations by engineered T cell therapy.

## Methods

**Ethics statement**. Frozen normal human TMA and frozen human lung cancer TMA were purchased from US Biomax (Rockville, USA). These deidentified samples are distributed by the facility under non-human subject study Institutional Review Board (IRB) approval. Therefore, the use of this material does not require IRB approval. Human PBMCs were purchased from the Gulf Coast Regional Blood Center (GCRBC, Houston, TX), and the donors' information was deidentified by GCRBC. Thus, the use of the PBMCs does not require IRB approval. All mouse experiments were performed in accordance with University of North Carolina (UNC) Animal Husbandry and Institutional Animal Care and Use Committee (IACUC) guidelines and were approved by UNC IACUC.

**Cell lines and cell culture**. Human NSCLC cell lines A549 (Cat# CCL-185), H1299 (Cat# CRL-5803), and SK-MES-1 (Cat# HTB-58) were originally purchased from American Type Culture Collection (ATCC); H460 (Cat# HTB-177) and H520 (Cat# HTB-182) were obtained from the Tissue Culture Facility of UNC at Chapel Hill, originally purchased from ATCC. All the cell lines were cultured in RPMI 1640 (Gibco) supplemented with 10% FBS and 2 mM GlutaMax. Penicillin (100 unit/mL) (Gibco) and streptomycin (100 μg/mL) (Gibco) were added to cell culture media. Cells were maintained in a humidified atmosphere containing 5% $CO_2$ at 37 °C. All cell lines were transduced with a retroviral vector encoding the GFP-Firefly-Luciferase (GFP-FFluc) gene[42]. Mycoplasma test is performed biweekly in the lab, all cell lines were mycoplasma free, and validated by flow cytometry for surface markers and functional readouts as needed.

**Human tissue samples**. Frozen normal human TMA and human NSCLC cancer TMA were purchased from US Biomax (Rockville, USA). The use of this material does not require IRB approval. These deidentified samples are distributed by the facility under non-human subject study IRB approval.

**Plasmid construction and retrovirus production**. The B7-H3.CAR construct was previously described[18]. Briefly, the B7-H3 specific scFv was derived from the 376.96 antibody[43,44] and the CAR used either the CD28 or 4-1BB costimulatory endodomains. The genes encoding CCR2a and CCR2b were amplified by PCR and linked together with the B7-H3.CAR construct using a 2A-like sequence[45]. The scFv specific for human CD19 was previously reported[46]. Retroviral supernatants

used for the transduction of human T cells were prepared as previously described[42]. Briefly, $2 \times 10^6$ 293T cells were seeded in 10 cm cell culture dish and transfected with the plasmid mixture of the retroviral vector, the Peg-Pam-e plasmid encoding MoMLV gag-pol, and the RDF plasmid encoding the RD114 envelope, using the GeneJuice transfection reagent (Merck Millipore), according to the manufacturer's instruction. Supernatant containing the retrovirus was collected 48 and 72 h after transfection, and filtered with 0.45 μm filters.

**Transduction and expansion of human T cells.** Human PBMCs, obtained from healthy volunteer donors (GCRBC, Houston, TX), were stimulated on plate-bound CD3 (1 ng/mL) and CD28 antibodies (1 ng/mL) (BD Biosciences, Mountain View, CA), in media containing 45% Click's media (Irvine Scientific, Santa Ana, CA), 45% RPMI 1640, 10% fetal bovine serum (Hyclone), 1% L-glutamine (Invitrogen, Carlsbad, CA), IL-7 (10 ng/mL) (PeproTech, Rocky Hill, NJ), and IL-15 (5 ng/mL) (PeproTech, Rocky Hill, NJ). Stimulated PBMCs were transduced with the retroviral vector encoding the CARs. Transduced cells were fed with IL-7 (10 ng/mL) and IL-15 (5 ng/mL) two-three times per week for 12–14 days of culture before subsequent analysis.

**Immunohistochemistry and tissue histopathology.** Tumor tissues were stained with the B7-H3 mAb overnight at 4 °C (clone 376.96, 1:1000 dilution, final concentration 1 μg/mL)[43], and then stained with HRP polymer conjugated goat anti-mouse secondary Ab (Dako, code K4000, 1:8 dilution) at 25 °C for 1.5 h. Slides were developed using DAB chromogen (Cell signaling), counterstained with CAT hematoxylin (Biocare medical), dehydrated in ethanol, and cleared in xylene (Fisher chemical). Cover slips were added using a histological mounting medium (Fisher, toluene solution). Stained TMA slides were digitally imaged at ×20 objective using the AperioScanScope XT (Leica). TMA slides were de-arrayed to visualize individual cores and each core was visually inspected. Folded tissues were excluded from the analysis using a negative pen, and all other artifacts were automatically excluded with the Aperio Genie software. The B7-H3 positive score was measured using Aperio membrane v9 (cell quantification) algorithm. Percentage of positive cells obtained with this algorithm at each intensity level (negative, low, medium, high) were used to calculate the H-Score using the formula: H-Score = (% at 1+) × 1+(% at 2+) × 2+(% at 3+) × 3. The Aperio color deconvolution v9 algorithm with the Genie classifier was also applied to calculate the area and intensity of the positive stain and generate a Score (0–300).

**ELISA.** CAR-T cells were cocultured with target tumor cells for 24 h before collecting the culture supernatant. The presence of IL2 and IFNγ was quantified by enzyme-linked immunosorbent assay (R&D Systems). ELISA data were collected with Synergy 2 Multi-Detection Microplate Reader with Gen5 v2.0.7 software (both BioTek).

**Flow cytometry.** The following Abs were purchased from BD bioscience: human CD3-APC-H7 (Cat#: 560176, dilution: 1:100); CD4-BV421 (Cat#: 562424, dilution: 1:100), CD8-APC (Cat#: 340584, dilution: 1:100), CD45-BV510 (Cat#: 563204, dilution: 1:30), CD45RA-PE (Cat#: 555489, dilution: 1:30), CCR7-FITC (Cat#: 561271, dilution: 1:50) and B7-H3-BV421 (Cat#: 565829, dilution: 1:100); human CCR2-BV421 (Cat#: 357210, dilution: 1:100) Ab was purchased from Biolegend. Expression of human B7-H3 in tumor cell lines and organoid cells was assessed with the 376.96 mAb followed by APC-goat anti-mouse IgG Ab (BD Biosciences, Cat# 550826, RRID: AB_398465, dilution 1:100), and confirmed with another B7-H3 mAb (Clone 7–517; BD Bioscience, Cat#: 565829, RRID: AB_2739369, dilution 1:100)[32]. Expression of the B7-H3.CARs was detected using the fusion protein 2Ig-B7-H3-Fc (R&D Cat# 1027-B3) and followed by AF647-goat anti-human IgG (H +L) Ab (Jackson ImmunoResearch Laboratories Inc., Cat# 109-606-088, dilution 1:100); CD19.CAR was detected with the previously described anti-CD19.CAR mAb[47]. Samples were acquired with BD FACS Canto II or BD FACS Fortessa using the BD Diva software (BD Biosciences). For each sample, a minimum of 10,000 events were acquired and data were analyzed using Flowjo 10.

**Coculture experiments.** Tumor cells were seeded in 24-well plates at a concentration of $5 \times 10^5$ cells/well. T cells were then added to the culture at a ratio of 1:5 for CAR-T cells to tumor cells without the addition of exogenous cytokines. Cells were analyzed on day 5 to measure residual tumor cells and T cells by FACS. Dead cells were gated out by Zombie Aqua Dye (Biolegend) staining, while T cells were identified by the expression of CD3 and tumor cells by the expression of GFP (NSCLC cell lines)[42,45]. For the coculture with organoids, the organoid LU6438B was cultured in an organoid culture medium before seeding to 48 well flat tissue culture plate. On the assay day, $5 \times 10^5$ organoid cells/well in 100 μL 5% MG were first seeded and CAR-T cells were added later at an E:T ratio of 1:2 or 1:5 in 1 mL RPMI plus 10% heat-inactivated FBS for coculture. Images of each condition were taken on day 1, and day 2, and cocultures were harvested for flow analysis on day 4. T cells were identified by the expression of CD45.

**Lung cancer organoid model establishment.** Organoid model LU6438B was established from PDX LU6438 model by Crown Bioscience Inc. In brief, LU6438

PDX tumors were expanded in mice and harvested when tumor volumes reached 200–800 mm³. Cleaned tumor tissues were minced, digested, and filtered to enrich fractions under 100 μm, which were used to establish organoids in the format of Matrigel suspension with an appropriate culture medium. Organoid size, stability, and expansion ability were monitored over time, as well as morphology by bright field microscopy. H&E staining and SNP analysis were applied to the established organoid model to validate its histopathological features and compare to the original tumor. All data can be accessed for free through Crownbio organoid database (https://organoid.crownbio.com).

**Proliferation assay.** T cells were labeled with 1.5 mM carboxyfluorescein diacetate succinimidyl ester (CFSE; Invitrogen) and plated with tumor cells at an E:T ratio of 1:2. CFSE dilution was measured on gated T cells on day 5 using flow cytometry[45].

**In vivo mouse studies.** All the mouse experiments were performed in accordance with UNC animal husbandry guidelines according to protocols approved by the UNC IACUC. Four- to six-week-old NSG female mice were purchased from the Animal Core Facility at the UNC at Chapel Hill, and housed in the Animal Core Facility at UNC until the age ready for the experiment (8–12 weeks old). Mice were maintained under specific-pathogen-free conditions with daily cycles of 12 h light–12 h darkness, temperature of 22 ± 2 °C, humidity of 55 ± 10, and continuous health monitoring was carried out on a regular basis. Animals were euthanized upon showing symptoms of clinically overt disease (not feeding, lack of activity, abnormal grooming behavior, hunched back posture) or excessive weight loss (15% body-weight loss over a week). For the A549 and H520 metastatic model, 8–10-week-old mice were injected intravenously via tail vein injection (i.v.) with $1 \times 10^6$ FFluc-A549 or $1.5 \times 10^6$ FFluc-H520 tumor cells. On day 14 after tumor cell inoculation, CD19.28, B7-H3.28 or B7-H3.BB CAR-T cells were injected i.v. ($5 \times 10^6$ cells/mouse). For the NSCLC orthotopic model, FFluc-A549 ($2 \times 10^5$) tumor cells were suspended in 50 μL Dulbecco's Phosphate-Buffered Saline and mixed with 50 μL Matrigel (corning), then surgically implanted into the left lung of 8–10-week-old mice as previously described[24,48]. Briefly, an incision was performed in the left thoracic cavity and NSCLC tumor cells were injected using a 28-gauge needle into the left lung. Seven days after tumor cell inoculation, CD19.28, B7-H3.28, or B7-H3BB CAR-T cells were injected i.v. ($10 \times 10^6$ cells/mouse). For the brain NSCLC tumor model, 10–12-week-old mice were injected intracranially (i.c.) with $5 \times 10^4$ FFluc-A549 cells or $4 \times 10^4$ FFluc-H520 cells in 3 μL of phosphate-buffered saline (PBS). The coordinates, with respect to the bregma, were 1 mm post, 3 mm right lateral, 3.5 mm deep, and within the nucleus caudatum. Upon tumor engraftment, $2 \times 10^6$ or $10 \times 10^6$ CD19.28, B7-H3.28 or CCR2b.B7-H3.28 CAR-T cells were injected i.v. via tail vein injection. For the lung orthotopic and brain co-xenograft model, the FFluc-H520 ($4 \times 10^4$) cells were implanted into the right hemisphere of the brain of 12-week-old NSG mice by i.c. injection. Five days later, $4 \times 10^5$ FFluc-H520 cells were injected into the left lung by intrathoracic injection, and 7 days later mice were treated with CD19.28, B7-H3.28, or CCR2b.B7-H3.28 CAR-T cells ($10 \times 10^6$ cells/mouse) by i.v. injection via tail vein. Investigators were not blinded, but mice were matched based on the tumor bioluminescence before assignment to control or treatment groups. Tumor growth in mice was monitored by bioluminescence imaging using the IVIS lumina II in vivo imaging system with Living Image software v4.5.2 (both PerkinElmer) or AMI Optical in vivo imaging system with Aura software v2.3.1 (both Spectral Instruments Imaging). Mice were euthanized in accordance with the institutional guidelines when signs of discomfort were detected by the investigators or as recommended by the veterinarian who monitored the mice three times a week. Since mouse death is not allowed as endpoint in our institution, we defined tumor BLI exceeding $3.5 \times 10^9$ photos/s as a surrogate for survival in these tumor models, in general, at this level of tumor BLI, mice start developing signs of sickness; the maximal tumor burden (BLI $3.5 \times 10^9$ photos/s) was not exceeded in all the mice experiments. To examine the CAR-T cell infiltration in the brain tumor model, 8 days post CAR-T cell infusion, mice were euthanized and brain was collected and dissociated using Human Tumor Dissociation Kit (Miltenyi Biotec) as per the manufacturer's instructions to generate a single-cell suspension for examining the infiltrating CAR-T cells by flow cytometry by staining CD45, CD3, CD4, and CD8.

**Chemotaxis transwell experiment.** T-cell chemotaxis was assayed by using 24-well Transwell chambers with 5 μm pores (Corning). A total of $2 \times 10^5$ cells in 100 μL chemotaxis buffer (RPMI 1640 with 10% FBS) were placed in the upper chambers. CCL2, diluted in 600 μL chemotaxis buffer was placed in the lower wells, as well as different cell culture supernatant, and the chambers were then incubated for 3 h in the incubator at 37 °C. Migrated cells located in the bottom wells were collected, washed once with PBS, and counted by FACS using CountBright™ Absolute Counting Beads (BD bioscience).

**Chemokine gene expression analysis.** Expression of CCL and CXCL family members (41 genes) were extracted from the summarized mRNA-seq V2 data of LUAD (n = 517) and LUSC (n = 501) cohorts available at the Genomic Data Commons Legacy Portal (https://portal.gdc.cancer.gov/legacy-archive). The CCL and CXCL expression data from MD Anderson cohort NSCLC primaries and paired brain metastasis (n = 31 subjects) were provided by Dr. Wistuba and

colleagues[20] as mean expression segregated by primary or brain metastasis. These expression estimates were generated from RNA of archival formalin-fixed and paraffin-embedded specimens assayed by Nanostring nCounter for 770 immune genes[20]. Expression data were log2 transformed and subjected to density estimation and plotted using RStudio v1.2.5. The quantile of CCL2 was estimated based on the expression distribution of all CCL and CXCL gene expression.

**Primers**. CCR2_NS_NcoI: CTCTAGACTGCCATGGAATCCACATCTCGTTCT CGGTTTATC; CCR2a-CAS-XhoI: CCGGATCGATCTCGAGCTAGGCTCCTTC TTTGTCCTGA; CCR2b- CAS-XhoI: CCGGATCGATCTCGAGTTATAAACCA GCCGAGACTTCCT; CCR2b_T2A_R: CACGTCCCCGCATGTTAGAAGACTT CCCCTGCCCTCTCCGCTTCCTAAACCAGCCGAGACTTCCTGCTC; T2A_ SP_F: GAAGTCTTCTAACATGCGGGGACGTGGAGGAAAATCCC GGGCCTA TGGAATTCGGCCTGAGCTGGCT.

**Quantification and statistical analysis**. The unpaired $t$ test with Welch's correction and two-tailed $p$ value calculation was used to measure the differences between the two groups. For multiple group comparisons, one-way ANOVA or two-way ANOVA was used to determine statistically significant differences between samples. $P$ values less than 0.05 were considered statistically significant. Measurements were summarized as mean ± SD as noted in the figure legends. The difference between the bioluminescence of the tumor and the survival curves was analyzed by the $\chi^2$ test using GraphPad Prism v8. Graph generation and statistical analyses were performed using the GraphPad Prism v8 software (GraphPad, La Jolla, CA).

**Reporting summary**. Further information on research design is available in the Nature Research Reporting Summary linked to this article.

## Data availability

The H&E staining and SNP analysis data of the lung cancer organoid models have been routinely generated when the organoid was initially established and can be accessed upon registration for free through Crownbio organoid database (https://organoid.crownbio. com). The chemokine gene expression data of NSCLC from TCGA (The Cancer Genome Atlas) are publicly available through the GDC (Genomic Data Commons) data portal (https://portal.gdc.cancer.gov). The CCL and CXCL expression data from MD Anderson cohort NSCLC primaries and paired brain metastasis were obtained confidentially from the corresponding author Dr Wistuba and the colleagues[20]. The original data set is not publicly available, access can be obtained by contacting Dr Wistuba[20]. The remaining data are available within the article, Supplementary information, or Source Data file. Source Data are provided with this paper.

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

## Acknowledgements

This work was supported in part University Research Fund (UCRF) (G.D.) and R21-CA229938-01A1 (G.D.) from National Cancer Institute. H.D. was supported by W81XWH-18-1-0441 from the Department of Defense (USA) and the Vicky Amidon Innovation Grant in Lung Cancer Research from the Lung Cancer Initiative of North Carolina. C.V.P. was supported in part by the National Institutes of Health (NIH) R01CA215075, the Jimmy V Foundation Scholar award, the UCRF Innovator Award, the Stuart Scott V Foundation/Lung Cancer Initiative Award for Clinical Research, the Lung Cancer Research Foundation, the Free to Breathe Metastasis Research Award and the Susan G. Komen Career Catalyst Award. E.B.H. was supported by a grant from the National Cancer Institute of the NIH under award number T32CA196589 and by the Lung Cancer Institute of North Carolina. The UNC Small Animal Imaging Facility at the Biomedical Imaging Research Center, and the Flow Cytometry Core Facilities are supported in part by an NCI Cancer Center Core Support Grant to the UNC Lineberger Comprehensive Cancer Center (P30-CA016086-40) United States. We thank Dr Wistuba who generously provided the RNA sequence data of the NSCLC primaries and paired brain metastasis.

## Author contributions

Conceptualization: H.D., Hongxia L., B.S., C.V.P., J.W., and G.D.; methodology: Hongxia L., H.D., E.B.H., J.C., Q.-X.L., J.G., J.S.P., B.S., C.V.P., and G.D.; investigation: Hongxia L., H.D., E.H., Huizhong L., K.H., J.C., Q.-X.L., J.P., C.V.P., B.S., and G.D.; writing—original draft: Hongxia L., H.D., and G.D.; supervision: H.D., B.S., C.V.P., and G.D.; review and editing: all authors.

## Competing interests

G.D. is a member of the scientific advisory board of Bellicum Pharmaceutical and Catamaran; G.D. and B.S. are consultants for Tessa Therapeutics; G.D. receives research support from Bluebird Bio and Bellicum Pharmaceutical; G.D. and H.D. filed a patent for the CAR targeting B7-H3, the patent number is US10519214, and the title is "Methods and Compositions for Chimeric Antigen Receptor Targeting Cancer Cells". No potential conflicts of interest were disclosed by the other authors.
