## [Peer Review File · Nature Communications]

Targeting Brain Lesions of Non-Small Cell Lung Cancer by Enhancing CCL2-Mediated CAR-T Cell MigrationREVIEWER COMMENTS

Reviewer #1 (Remarks to the Author): with expertise in lung cancer and CAR-T

Authors address an important clinical problem of treating brain metastases from NSCLC, a common and vexing problem as many agents including T cells are impaired in crossing the BBB.

They conduct experiments to take advantage of the chemokine gradient resulting from tumor cells, by treating mice with brain tumors with CCR2b CARs targeted against B7-H3 antigen.

They conclude that CCR2b CARs pass the blood brain barrier and treat brain metastases effectively.

Novelty: Designing CARs to respond better to a single tumor-secreted chemokine gradient has been published as authors refer to. This study adds to another such example as a proof-of-principle study. Addressing BBB and brain metastases by CAR T cells is novel, however, the data presented do not justify the conclusions (please see below). No where in the manuscript BBB is addressed except an indirect evidence of CAR T-cell accumulation in brain tumor (please see below). Author should try and address below critique to convince that conclusions can be justified.

Critique:

1. While there are increased # of CAR T cells with CCR2b compared to CAR T cells alone, is this because of chemokine gradient in the tumor? Do authors have percentage / number of CAR and non-CAR T cells in the tumor and normal brain away from the tumor? How one can conclude that this is tumor specific accumulation?
2. If it is indeed tumor-specific accumulation, how come the CCR2b CAR T cells fail to show efficacy in tail vein model as these tumors also express chemokine and create gradient.
3. While authors mention that CAR T cells were ineffective in brain tumor model (S Fig 2E-F), in experiments with CCR2b CAR T cells, CAR t cells alone show prolonged median survival to 60-70 days compared to 20 days in control mice (Fig 4D, S Fig 6D). S Fig 2E-F do not show any correlative data (BLI, survival, weights etc) and simply claim that CAR T cells are not effective.
4. Mice with intracranial brain tumor (tumor, not metastases) show increased accumulation of CAR T cells when co-expressed with CCR2b compared to CARs alone. However, these mice do not have any other sites of disease to increase the proliferation of CAR T cells like in patients. Hence the model is artificial. I do understand that mice survival may be an issue with multiple sites of tumor, but this limitation needs to be discussed in the discussion section.
5. In orthotopic model experiment (Fig 2E-H), tumor seems to be intrapleural than orthotopic by interpretation of BLI. Although there may be some spillage of photon elision in BLI figures, typically orthotopic tumor show high BLI in left lung with mediastinal lymph node metastases.
6. The presented data of high CCL2 in brain compared to primary lung show slight increase, seem to be subjective determination than any statistical interpretation.
7. What is the normal expression of B7-H3 in brain tissue? What is the normal expression of CCL2 in brain tissue? What percentage of brain tumor cells express both B7-H3 and CCL2 or at least high expression of B7-H3 or CCL2? Without co-expression, this proof-of-principle study is not translationally relevant.
8. Pertaining to the above question, what is the expected toxicity? Authors should atleast try to attempt to address this question by discussing antigen expression gradient between normal and tumor tissue.
9. The doses used across experiments vary between 3, 4 and 10 million CAR T cells making it difficult to interpret the data across experiments.
10. The CAR T-cell dose use din majority of the experiments is 10 million CAR T cells per NSG immunodeficient mouse (30 gm mouse). This is equal to 4 billion CAR T cells per 70 kg immunocompetent patient bearing immunosuppressive tumor. Please provide some discussion regarding dosage in the discussion section as the study aims to be translational.
11. As written, the discussion is a mere reproduction of results with no reflection of above-mentioned questions.

Reviewer #2 (Remarks to the Author): with expertise in CAR-T and brain cancer

The authors describe 2 relatively new findings in this manuscript: (1) that NSCLC cell lines, organoids, and PDX/orthotopic and metastatic models express and can be targeted with B7-H3-directed CAR T cells, and (2) that co-expression of CCR2b in the CAR T cells improves responses and trafficking of CAR T cells in intracranial tumor deposits. The second point has been demonstrated before in other tumor models (Craddock PMID 20842059, in GD2 CARs, which is referenced, and Moon PMID 21610146, in mesothelin CARs, which is not reference but should be). This diminishes enthusiasm for the the second novelty point, but the use of the various NSCLC models and the new target do add to the scientific community. I also note that B7-H3 CARs have already been described extensively by this group, in glioblastoma.

I have only minor points to make:

1. Can you clarify the source of the B7-H3 antibody that is used to detect expression. There is a reference, but there should be some reference as to how it is obtained or whether it is commercially available.
2. For the transwell experiment, please describe in the Results (page 12) that it was a transwell and what the pore size was rather than just saying "migration." It is different to migrate across a transwell than an endothelial cell.
3. Figure 4A -- the schematic says "IT" but the legend says "intracranially IC". It looks like the schematic needs to be changed to IC.

Reviewer #3 (Remarks to the Author): with expertise in CAR-T and lung cancer

In this paper, Li et al. demonstrate that B7-H3 is an attractive CAR target for NSCLC, as B7-H3-targeted CAR-T cells show substantial anti-tumor activity against NSCLC cell lines and organoids in vitro and in orthotopic and metastatic models of NSCLC in vivo. To address the challenge of treating NSCLC that has metastasized to the brain, Li et al. show that the chemokine Ccl2 is highly expressed in primary NSCLC and matched brain metastases, and that CAR-T cells engineered to co-express the cognate chemokine receptor CCR2b show improved migration towards NSCLC cell lines in vitro and improved control of NSCLC tumors in a brain metastasis model in vivo. Overall, the studies are well-powered and well-executed, and the in vivo results in the intracranial model are particularly striking.

B7-H3 expression in NSCLC has been previously described (PMID: 28539467 among others), and the in vivo efficacy of B7-H3 CAR-T cells has also been previously shown in xenograft models of NSCLC using subcutaneous or intravenous administration of A549 tumor cells (PMID: 32002297; Liu et al J of Hematology & Oncology 2021). Likewise, expression of CCR2b as a strategy to improve trafficking of CAR-T cells towards neuroblastoma tumors was also shown, albeit in a subcutaneous xenograft model (PMID: 20842059). Thus, the main novelty of this study lies in the development of an intracranial model of brain metastasis to interrogate whether the CCR2b-overexpression strategy is sufficient to enable CAR-T cells to penetrate the brain. The inability of CAR-T cells to access non-lymphoid tissues efficiently is a major barrier to the success of this therapy in solid tumors, and the brain is one of the least accessible sites to intravenously infused T cells, with most brain tumors requiring intracranial infusion of T cells to achieve any efficacy. Thus, strategies to enhance access of CAR-T cells to solid tumors, and to the brain in particular, are severely lacking. Although CCR2b was shown to improve trafficking of CAR-T cells to neuroblastoma tumors implanted subcutaneously in a xenograft model (PMID: 20842059), whether this strategy was sufficient to enable CAR-T cells to cross the blood-brain barrier, a much more formidable obstacle, has not been previously demonstrated. Li et al.'s results showing enhanced control of NSCLC tumors in the brain by CCR2b CAR-T cells are convincing and provide evidence that CCR2b expression is a viable strategy to enhance access of CAR-T cells to the brain.

Minor comments for revision:

Fig. 1B: If 300 is the maximum positive score possible, the scale of the y-axis should be changed to extend to 300 to make it clearer where LUAD and LUSC scoring lies on the positivity scale.

Fig. S2: What does it mean that mice were "censored" when BLI reached a certain limit? Was BLI signal exceeding the 3.5×10^9 photons/s limit considered an endpoint criteria and surrogate for survival?

Fig. 3H: Does CAR MFI differ between the different constructs?

Fig. 4E: The only data tying increased activity of CCR2b CAR-T cells to increased trafficking in vivo is the right panel in Fig. 4E, where CCR2b CAR-T cell numbers are only slightly increased in the brain, though this could be confounded by reduced tumor burden/antigen in this group at this time point (based on BLI in Fig. 4B). Are CAR-T cell numbers increased at earlier time points when tumor burden is still similar?

Most patients with brain metastases also carry primary lung tumors: is the migration of B7-H3 CAR-T cells to the brain impacted by the presence of primary lung tumors? As both primary lung tumors and brain mets are predicted to produce CCL2, would a CCL2 "sink" in the lung interfere with the ability of CAR-T cells to efficiently access and control the growth of brain mets?

Reviewer #1: with expertise in lung cancer and CAR-T

Remarks to the Author:

Authors address an important clinical problem of treating brain metastases from NSCLC, a common and vexing problem as many agents including T cells are impaired in crossing the BBB. They conduct experiments to take advantage of the chemokine gradient resulting from tumor cells, by treating mice with brain tumors with CCR2b CARs targeted against B7-H3 antigen. They conclude that CCR2b CARs pass the blood brain barrier and treat brain metastases effectively.

Novelty: Designing CARs to respond better to a single tumor-secreted chemokine gradient has been published as authors refer to. This study adds to another such example as a proof-of-principle study. Addressing BBB and brain metastases by CAR T cells is novel, however, the data presented do not justify the conclusions (please see below). Nowhere in the manuscript BBB is addressed except an indirect evidence of CAR T-cell accumulation in brain tumor (please see below). Author should try and address below critique to convince that conclusions can be justified.

Critique:

1. While there are increased # of CAR T cells with CCR2b compared to CAR T cells alone, is this because of chemokine gradient in the tumor? Do authors have percentage / number of CAR and non-CAR T cells in the tumor and normal brain away from the tumor? How one can conclude that this is tumor specific accumulation?

We conducted additional experiments to better define that the increased CAR-T cell number in the brain of CCR2b co-expressed CAR-T cell treated mice is tumor specific. We implanted A549 tumor cells into the right brain hemisphere in NSG mice, and 18 days later mice were treated with B7-H3.CAR-Ts or CCR2b.B7-H3.CAR-Ts via i.v. injection. Eight days after CAR-T cell treatment, mice were euthanized and T cells in the blood, spleen, left brain hemisphere (without tumor) and right brain hemisphere (with tumor) were quantified by flow cytometry. These new data are summarized in the new **Fig. 5F**. In the left brain hemisphere (without tumor), we did not observe any increase of T-cell numbers in mice receiving CCR2b.B7-H3.CAR-Ts compared to those receiving B7-H3.CAR-Ts. In contrast, increased T cells numbers were consistently observed in the right brain hemisphere (with tumor) of CCR2b.B7-H3.CAR-Ts treated mice compared to those treated with B7-H3.CAR-Ts (**Fig. 5F**). These data support the claim that the increased number of CAR-T cells with CCR2b is tumor specific.

Due to the technical difficulties, we cannot quantify how many human T cells detected in the brain are CAR positive. We used the B7-H3-Fc protein to detect the CAR expression in T cells¹. Unfortunately, this strategy cannot be reliably used to detect CAR expression in T cells localized in the tumor, since the CAR epitope is masked by the B7-H3 expressed by tumor cells. The CAR is detectable in CAR-Ts circulating in the blood using the B7-H3-Fc protein-based staining, but the staining is very dim in T cells detected within the tumor.

New Figure 5F. In the separate experiment, mice were euthanized 8 days after CAR-T cell infusion, human CD45⁺CD3⁺ T cells were enumerated in blood, spleen, and both left brain hemisphere (without tumor) and right brain hemisphere (with tumor) by flow cytometry. Bar graph summary are shown (n=5 mice/group), **p = 0.0062, ****p < 0.0001, two-way ANOVA with Sidak's multiple comparisons test.

2. If it is indeed tumor-specific accumulation, how come the CCR2b CAR T cells fail to show efficacy in tail vein model as these tumors also express chemokine and create gradient.

We only observed modest enhanced antitumor activity for the CCR2b co-expressing CAR-T cells in the metastatic model. We think that in this model in which tumor cells engraft mostly in the liver, chemokine gradients may play a limited role in governing the trafficking of CAR-T cells within the liver when the CAR-T cells are infused intravenously. We have included a comment in the discussion of the manuscript.

3. While authors mention that CAR T cells were ineffective in brain tumor model (S Fig 2E-F), in experiments with CCR2b CAR T cells, CAR T cells alone show prolonged median survival to 60-70 days compared to 20 days in control mice (Fig 4D, S Fig 6D). S Fig 2E-F do not show any correlative data (BLI, survival, weights etc) and simply claim that CAR T cells are not effective.

We apologize for the error of labeling the Fig. S2E. The actual dose of CAR-T cells used in this experiment is 2×10^6 /mouse and not 10×10^6 /mouse. This error was caused by the copy and paste of the schema from other figures. The number of CAR-T cells used in Fig. 4D (in the revised new version of figures is Fig. 5D) and Fig. S6D (in the revised new version of supplementary figures is Fig. S7D) is 10×10^6 /mouse, therefore the B7-H3-CAR-Ts (without CCR2b) showed better antitumor activity compared to that in Fig. S2F. However, due to the small sample size, we did not show the survival curve or tumor BLI for the Fig S2E-F in the initial submission. We conducted additional experiments and treated the mice with brain tumor (A549) with two different doses of CAR-T cells (2×10^6 /mouse or 10×10^6 /mouse). As showed in the new Fig S3A-D, we did not observe significant antitumor effects in mice treated with 2×10^6 B7-H3.CAR-Ts, but significant antitumor effects in mice treated with 10×10^6 B7-H3.CAR-Ts (new Fig S3E-H). These data are consistent with the data previously showed in Fig. 4D (in the revised new version of figures is Fig. 5D) and Fig. S6D (in the revised new version of supplementary figures is Fig. S7D)

Figure S3. B7-H3.CAR-Ts cannot efficiently control the A549 tumor growth in the brain. (A) Schematic of the A549 brain tumor model in which NSG mice are implanted i.c. with the FFluc-A549 cells (5×10^4 cells) and treated with low dose CAR-T cells (2×10^6) 16 days after tumor implantation by i.v. injection. (B and C) Representative tumor BLI images (B) and BLI kinetics (C) of the A549 tumor growth in the model shown in (A). (D) Kaplan-Meier survival curves of the mice in (B), $n = 4$ mice/group. (E) Schematic of the A549 brain tumor model in which NSG mice are implanted i.c. with FFluc-A549 cells (5×10^4 cells) and treated with high dose CAR-T cells (10×10^6) 17 days after tumor implantation by i.v. injection. (F and G) Representative tumor BLI images (F) and BLI kinetics (G) of the A549 tumor growth in the model shown in (E). (H) Kaplan-Meier survival curves of the mice in (E), $**p = 0.0045$, chi-square test, $n = 5$ mice/group. Days indicated in (B, F) represent the days post T-cell infusion.

4. Mice with intracranial brain tumor (tumor, not metastases) show increased accumulation of CAR T cells when co-expressed with CCR2b compared to CARs alone. However, these mice do not have any other sites of disease to increase the proliferation of CAR T cells like in patients. Hence the model is artificial. I do understand that mice survival may be an issue with multiple sites of tumor, but this limitation needs to be discussed in the discussion section.

We appreciate that the reviewer acknowledges the technical difficulties in performing experiments *in vivo* in which the mice are simultaneously presenting tumor lesions in the lung and in the brain. However, since this is an important point, we attempted performing these experiments upon the approval of the mouse protocol in which two surgical procedures are allowed to establish tumor localization in both lung and brain (new **Figure 6A**). We treated the mice with CD19.CAR-Ts, B7-H3.CAR-Ts or CCR2b.B7-H3.CAR-Ts via i.v. inoculation. As illustrated in the new **Figure 6B-D**, we found that the CCR2b.B7-H3.CAR-Ts showed superior antitumor activity against the tumor lesions in the brain, especially at day 10 after CAR-T cell treatment (**Figure 6B**). These data indicate that the co-expression of CCR2b enhances the migration of CAR-T cells towards the brain tumor and better controls the tumor growth in the brain even in the condition of tumor lesions simultaneously present in the lung.

Figure 6. CCR2b co-expressing B7-H3.CAR-Ts show superior antitumor activity in the H520 lung orthotopic and brain co-xenograft tumor model. (A) Schematic of the H520 lung orthotopic and brain co-xenograft tumor model in NSG mice. Briefly, the FFluc-H520 (4×10^4) cells were implanted into the right hemisphere of the brain of NSG mice by i.c. injection, then 5 days later, 4×10^5 FFluc-H520 cells were implanted into the left lung by intrathoracic injection. Seven days later, mice were treated with CD19.28-CAR-Ts, B7-H3.28-CAR-Ts or CCR2b.B7-H3.28-CAR-Ts (10×10^6 cells/mouse) by i.v. injection. (B) Representative tumor BLI images. Days represent the days post T-cell infusion. (C) The BLI kinetics of the FFluc-H520 tumor growth in the model shown in (B), BLI signals from brain (top) and whole body (bottom) are illustrated, respectively. (D) Kaplan-Meier survival curves of mice in (B), $n = 5$ mice/group, $*p = 0.0486$ for B7-H3.28 vs. CCR2b.B7-H3.28, $*p = 0.0181$ for CD19.28 vs. B7-H3.28, $**p = 0.0026$ for CD19.28 vs. CCR2b.B7-H3.28, chi-square test. ($n = 5$).

5. In orthotopic model experiment (Fig 2E-H), tumor seems to be intrapleural than orthotopic by interpretation of BLI. Although there may be some spillage of photon elision in BLI figures, typically orthotopic tumor show high BLI in left lung with mediastinal lymph node metastases.

As indicated in the methodology section, the lung orthotopic model was conducted by injecting the tumor cells mixed with Matrigel into the parenchyma of the left lung by intra-thoracic injection. We cannot exclude the possibility that the tumor cells will also engraft in the pleural cavity. However, the tumor BLI images seem indicating that most of the mice have higher BLI signal in the left lung than in the right lung, despite the fact that BLI cannot fully discriminate the anatomic location of the tumor. In mice analyzed by necropsy, tumors were detectable macroscopically in the lung parenchyma. Liver metastases were also detectable at later time points. We did not look for the tumor metastasis to lymph nodes in this model due to the difficulties to find lymphonodes in NSG mice.

6. The presented data of high CCL2 in brain compared to primary lung show slight increase, seem to be subjective determination than any statistical interpretation.

Kudo et al. provided summarized data as reported in **Table S1**². Although we have contacted the corresponding author to request the original data, they declined the request so we report the data as previously published.

7. What is the normal expression of B7-H3 in brain tissue? What is the normal expression of CCL2 in brain tissue? What percentage of brain tumor cells express both B7-H3 and CCL2 or at least high expression of B7-H3 or CCL2? Without co-expression, this proof-of-principle study is not translationally relevant.

We and others, have previously reported that B7-H3 is highly expressed in glioblastoma and other pediatric brain malignancies, while normal brain does not show B7-H3 expression^{1,3-5}. To explore the CCL2 expression in normal brain tissue, we analyzed the CCL2 expression data in 51 different normal tissues from the GTEx (Genotype-Tissue Expression) project, and we found that brain has the lowest level for CCL2 expression across all tissues studied.

As indicated in Question-6, the authors of Kudo et al. only provided summarized data as group means. Therefore we could not do further explore what percentage of brain tumor cells express both B7-H3 and CCL2.

8. Pertaining to the above question, what is the expected toxicity? Authors should at least try to attempt to address this question by discussing antigen expression gradient between normal and tumor tissue.

As mentioned earlier, we and others have reported that B7-H3 is not express in normal brain¹⁻⁴. Therefore, we do not expect any obvious toxicity for B7-H3 CAR-T cell treatment or increased toxicity by co-expressing CCR2 with CAR-T cells. A recent case report in which B7-H3-CAR-T cells were infused in a patient with meningioma did not report detectable toxicity⁶. We have further discussed that in the Discussion section of the manuscript.

9. The doses used across experiments vary between 3, 4 and 10 million CAR T cells making it difficult to interpret the data across experiments.

The different tumor cell lines have different growth rate. Therefore, for each model, we have tested different doses of the CAR-T cells in different experiments, and what we showed in the manuscript are the CAR-T cell number needed to achieve antitumor activity in each model.

10. The CAR T-cell dose use in majority of the experiments is 10 million CAR T cells per NSG immunodeficient mouse (30 gm mouse). This is equal to 4 billion CAR T cells per 70 kg immunocompetent patient bearing immunosuppressive tumor. Please provide some discussion regarding dosage in the discussion section as the study aims to be translational.

The literature of preclinical models of CAR-T cell therapies acknowledges the limitations of these models. The goal of these models is not to suggest the dose of T cells that should be used in clinical studies. A dose escalation of CAR-T cells ranging from 1×10^6 to 1×10^9 /kg is frequently used in Phase I clinical studies to define the maximum tolerated dose.

11. As written, the discussion is a mere reproduction of results with no reflection of above-mentioned questions.

We have included modifications to the discussion.

Reviewer #2: with expertise in CAR-T and brain cancer

Remarks to the Author:

The authors describe 2 relatively new findings in this manuscript: (1) that NSCLC cell lines, organoids, and PDX/orthotopic and metastatic models express and can be targeted with B7-H3-directed CAR T cells, and (2) that co-expression of CCR2b in the CAR T cells improves responses and trafficking of CAR T cells in intracranial tumor deposits. The second point has been demonstrated before in other tumor models (Craddock PMID 20842059, in GD2 CARs, which is referenced, and Moon PMID 21610146, in mesothelin CARs, which is not reference but should be). This diminishes enthusiasm for the second novelty point, but the use of the various NSCLC models and the new target do add to the scientific community. I also note that B7-H3 CARs have already been described extensively by this group, in glioblastoma.

We added the suggested reference.

I have only minor points to make:

1. Can you clarify the source of the B7-H3 antibody that is used to detect expression. There is a reference, but there should be some reference as to how it is obtained or whether it is commercially available.

We clarify the origin of the B7-H3.CAR and the Ab used to generate the CAR in the method section.

2. For the transwell experiment, please describe in the Results (page 12) that it was a transwell and what the pore size was rather than just saying "migration." It is different to migrate across a transwell than an endothelial cell.

We added the information in the method section and in the figure legend. The size of the pores of the transwell chambers are 5 μ m.

3. Figure 4A -- the schematic says "IT" but the legend says "intracranially IC". It looks like the schematic needs to be changed to IC.

We apologize for the typo. We corrected it to i.c..

Reviewer #3: with expertise in CAR-T and lung cancer

Remarks to the Author:

In this paper, Li et al. demonstrate that B7-H3 is an attractive CAR target for NSCLC, as B7-H3-targeted CAR-T cells show substantial anti-tumor activity against NSCLC cell lines and organoids in vitro and in orthotopic and metastatic models of NSCLC in vivo. To address the challenge of treating NSCLC that has metastasized to the brain, Li et al. show that the chemokine Ccl2 is highly expressed in primary NSCLC and matched brain metastases, and that CAR-T cells engineered to co-express the cognate chemokine receptor CCR2b show improved migration towards NSCLC cell lines in vitro and improved control of NSCLC tumors in a brain metastasis model in vivo. Overall, the studies are well-powered and well-executed, and the in vivo results in the intracranial model are particularly striking.

B7-H3 expression in NSCLC has been previously described (PMID: 28539467 among others), and the in vivo efficacy of B7-H3 CAR-T cells has also been previously shown in xenograft models of NSCLC using subcutaneous or intravenous administration of A549 tumor cells (PMID: 32002297; Liu et al J of Hematology & Oncology 2021). Likewise, expression of CCR2b as a strategy to improve trafficking of CAR-T cells towards neuroblastoma tumors was also shown, albeit in a subcutaneous xenograft model (PMID: 20842059). Thus, the main novelty of this study lies in the development of an intracranial model of brain metastasis to interrogate whether the CCR2b-overexpression strategy is sufficient to enable CAR-T cells to penetrate the brain. The inability of CAR-T cells to access non-lymphoid tissues efficiently is a major barrier to the success of this therapy in solid tumors, and the brain is one of the least accessible sites to intravenously infused T cells, with most brain tumors requiring intracranial infusion of T cells to achieve any efficacy. Thus, strategies to enhance access of CAR-T cells to solid tumors, and to the brain in particular, are severely lacking. Although CCR2b was shown to improve trafficking of CAR-T cells to neuroblastoma tumors implanted subcutaneously in a xenograft model (PMID: 20842059), whether this strategy was sufficient to enable CAR-T cells to cross the blood-brain barrier, a much more formidable obstacle, has not been previously demonstrated. Li et al.'s results showing enhanced control of NSCLC tumors in the brain by CCR2b CAR-T cells are convincing and provide evidence that CCR2b expression is a viable strategy to enhance access of CAR-T cells to the brain.

Minor comments for revision:

1. Fig. 1B: If 300 is the maximum positive score possible, the scale of the y-axis should be changed to extend to 300 to make it clearer where LUAD and LUSC scoring lies on the positivity scale.

As the reviewer suggested, we adjusted the maximum value of y-axis from 200 to 250.

Figure 1B. Summary of B7-H3 expression score of the immunochemistry (IHC) results in normal lung and NSCLC.

2. Fig. S2: What does it mean that mice were “censored” when BLI reached a certain limit? Was BLI signal exceeding the 3.5×10^9 photos/s limit considered an endpoint criteria and surrogate for survival?

Mouse death is not allowed in our institution as endo point of mouse experiments. Therefore, we defined tumor BLI exceeding 3.5×10^9 photos/s as the endpoint and surrogate for survival if the mouse is still alive, now we removed this sentence into method section from the figure legend. In general, at this level of tumor BLI, mice start developing signs of sickness, such as swollen belly and decreased activity.

3. Fig. 3H: Does CAR MFI differ between the different constructs?

Fig. 3H is moved to Fig. 4A in the new revised version of figures. As suggested by the reviewer, we show the MFI of the CAR in the revised **Figure 4C**. CCR2b co-expression slightly decreased the MFI of CAR, but without impairing the overall transduction efficiency of CAR-T cells (**Figure 4A-C**).

Figure 4a-c. (a) Representative flow plots of CARs expression in CD19.28, B7-H3.28, B7-H3.BB, CCR2b.B7-H3.28 and CCR2b.B7-H3.BB CAR-T cells as assessed by flow cytometry. (b, c) Summary of the percentage (b) and MFI (c) of CAR expression in B7-H3.28, B7-H3.BB, CCR2b.B7-H3.28 and CCR2b.B7-H3.BB CAR-T cells (n = 10 independent experiments using CAR-T cells generated from 10 different donors, *p = 0.0219 for CD19.28 vs. B7-H3.28, *p = 0.0240 for CD19.28 vs. B7-H3.BB, *p = 0.0295 for B7-H3.BB vs. CCR2b.B7-H3.28, *p = 0.0217 for B7-H3.BB vs. CCR2b.B7-H3.BB, one-way ANOVA). Data are presented as mean values +/- SD.

4. Fig. 4E: The only data tying increased activity of CCR2b CAR-T cells to increased trafficking in vivo is the right panel in Fig. 4E, where CCR2b CAR-T cell numbers are only slightly increased in the brain, though this could be confounded by reduced tumor burden/antigen in this group at this time point (based on BLI in Fig. 4B). Are CAR-T cell numbers increased at earlier time points when tumor burden is still similar?

Fig. 4E is Fig. 5E in the new revised version of figures. We tried to examine the T-cell numbers at early time points, such as day 3 and day 5 after CAR-T cell infusion. However, few CAR-T cells can be clearly detected in the brain in all groups at this time point, and we cannot exclude that this is caused by technical difficulties in enumerating T cells from tissue single cell suspension and flow cytometry analysis. However, a good number of T cells can be detected at day 8, which is likely the results of both migration and subsequent proliferation of the T cells upon encountering the tumor.

5. Most patients with brain metastases also carry primary lung tumors: is the migration of B7-H3 CAR-T cells to the brain impacted by the presence of primary lung tumors? As both primary lung tumors and brain mets are predicted to produce CCL2, would a CCL2 “sink” in the lung interfere with the ability of CAR-T cells to efficiently access and control the growth of brain mets?

As indicated in question #4 of reviewer #1, we have established the lung orthotopic and brain co-xenograft model in NSG mice (new **Figure 6A**), and then treated the mice with CD19.CAR-Ts, B7-H3.CAR-Ts or CCR2b.B7-H3.CAR-Ts by i.v. infusion. As illustrated in the new **Figure 6B-D**, we found that the CCR2b.B7-H3.CAR-Ts showed superior antitumor activity against the tumor lesions in the brain, especially at day 10 after CAR-T cell treatment (**Figure 6B**). These data indicate that the co-expression of CCR2b enhances the migration of CAR-T cells towards the brain tumor and better controls the tumor growth in the brain even in the condition of tumor lesions simultaneously present in the lung.

Figure 6. CCR2b co-expressing B7-H3.CAR-Ts show superior antitumor activity in the H520 lung orthotopic and brain co-xenograft tumor model. (A) Schematic of the H520 lung orthotopic and brain co-xenograft tumor model in NSG mice. Briefly, the FFluc-H520 (4×10^4) cells were implanted into the right hemisphere of the brain of NSG mice by i.c. injection, then 5 days later, 4×10^5 FFluc-H520 cells were implanted into the left lung by intrathoracic injection. Seven days later, mice were treated with CD19.28-CAR-Ts, B7-H3.28-CAR-Ts or CCR2b.B7-H3.28-CAR-Ts (10×10^6 cells/mouse) by i.v. injection. (B) Representative tumor BLI images. Days represent the days post T-cell infusion. (C) The BLI kinetics of the FFluc-H520 tumor growth in the model shown in (B), BLI signals from brain (top) and whole body (bottom) are illustrated, respectively. (D) Kaplan-Meier survival curves of mice in (B), $n = 5$ mice/group, * $p = 0.0486$ for B7-H3.28 vs. CCR2b.B7-H3.28, * $p = 0.0181$ for CD19.28 vs. B7-H3.28, ** $p = 0.0026$ for CD19.28 vs. CCR2b.B7-H3.28, chi-square test. ($n = 5$).

Reference List

1. Du H, Hirabayashi K, Ahn S et al. Antitumor Responses in the Absence of Toxicity in Solid Tumors by Targeting B7-H3 via Chimeric Antigen Receptor T Cells. *Cancer Cell* 2019;35:221-237.

2. Kudo Y, Haymaker C, Zhang J et al. Suppressed immune microenvironment and repertoire in brain metastases from patients with resected non-small-cell lung cancer. *Ann.Oncol.* 2019;30:1521-1530.
3. Nehama D, Di IN, Musio S et al. B7-H3-redirected chimeric antigen receptor T cells target glioblastoma and neurospheres. *EBioMedicine.* 2019;47:33-43.
4. Picarda E, Ohaegbulam KC, Zang X. Molecular Pathways: Targeting B7-H3 (CD276) for Human Cancer Immunotherapy. *Clin.Cancer Res.* 2016;22:3425-3431.
5. Majzner RG, Theruvath JL, Nellan A et al. CAR T Cells Targeting B7-H3, a Pan-Cancer Antigen, Demonstrate Potent Preclinical Activity Against Pediatric Solid Tumors and Brain Tumors. *Clin Cancer Res.* 2019;25:2560-2574.
6. Tang X, Liu F, Liu Z et al. Bioactivity and safety of B7-H3-targeted chimeric antigen receptor T cells against anaplastic meningioma. *Clin Transl.Immunology* 2020;9:e1137.

REVIEWERS' COMMENTS

Reviewer #1 (Remarks to the Author):

Authors addressed various questions in a reasonable fashion with additional experiments and data and provided fair responses to unanswered questions.
The additional experiments provide satisfactory data.

Reviewer #3 (Remarks to the Author):

Thank you for addressing my earlier comments.